# Two-dimensional fetal speckle tracking; a learning curve study for offline strain analysis

Chantelle M. de Vet[1,2,3]*, Thomas J. Nichting[1,2,3], Annemarie F. Fransen[1], Daisy A. A. van der Woude[1], Myrthe van der Ven[1,3,4], Richard A. J. Post[5], Zoé van Lier[1], S. Guid Oei[1,2,3], Noortje H. M. van Oostrum[6], Judith O. E. H. van Laar[1,2,3]

1 Department of Gynecology and Obstetrics, Máxima Medical Center, Veldhoven, Noord-Brabant, The Netherlands, 2 Department of Electrical Engineering, Eindhoven University of Technology, Eindhoven, Noord-Brabant, The Netherlands, 3 Eindhoven MedTech Innovation Center, Eindhoven, Noord-Brabant, The Netherlands, 4 Department of Biomedical Engineering, Eindhoven University of Technology, Eindhoven, Noord-Brabant, The Netherlands, 5 Department of Mathematics and Computer Science, Eindhoven University of Technology, Eindhoven, Noord-Brabant, The Netherlands, 6 Department of Gynecology and Obstetrics, Ghent University Hospital, Gent, Oost-Vlaanderen, Belgium

☉ These authors contributed equally to this work.
* chantelle.de.vet@mmc.nl, c.m.vet@tue.nl

**Data Availability Statement:** All relevant data are within the paper and its Supporting Information files.

## Abstract

### Objectives

Two-dimensional speckle tracking (2D-STE) strain analysis holds promise for assessing fetal cardiac function. Understand the learning curve before introducing 2D-STE into obstetrics is crucial. This study examined the learning curve for offline analysis of fetal left (LV) and right ventricular (RV) global longitudinal strain (GLS) using 2D-STE.

### Methods

After 2D-STE training, three trainees (Maternal-Fetal Medicine fellow, OBGYN resident and medical student) analyzed 100 fetal heart clips using 2D-STE to calculate LV- and RV-GLS. Intra-class correlation coefficients (ICC) and Bland-Altman plots were compared GLS values across four sets of 25 clips for each trainee against the expert. Repeated measurements analysis compared GLS score differences between expert and trainees over time and among trainees, adjusting p-values with a Bonferroni correction.

### Results

LV-GLS consistency evolved from poor-to-moderate during the first 50 measurements to moderate-to-good during the second 50 for all trainees. RV-GLS consistency evolved from poor-to-moderate during the first 75 measurements to moderate-to-good during the final 25 measurements for the fellow and resident. The student's RV-GLS consistency was poor during the first 25 measurements, moderate-to good during the second 25 measurements and again poor-to-moderate during the final 50 measurements. Repeated measurements analysis showed a significant decrease in variability of the LV- and RV-GLS score differences between the expert and trainees over time ($p_{adj}$<0.001), which was not significantly different between trainees. Moreover, the mean of those differences were significantly

**Funding:** The author(s) received no specific funding for this work.

**Competing interests:** The authors have declared that no competing interests exist.

**Abbreviations:** 2D-STE, Two-dimensional speckle tracking echocardiography; 95%, CI 95% confidence intervals; GLS, Global Longitudinal Strain; LV, Left ventricular; RV, Right ventricular.

different for all trainees for LV-GLS ($p_{adj}$<0.001) and RV-GLS ($p_{adj}$ = 0.029), and did significantly change over time for RV-GLS ($p_{adj}$<0.001) but not for LV-GLS.

## Conclusions

A clear learning effect was observed by the significant decrease in variability of the difference in the score between the expert and trainees over time. The consistency of fetal GLS analysis with 2D-STE was generally found to be moderate to good after 100 measurements in trainees.

## Introduction

The fetal heart is a central organ in the adaptation to physiological and pathological changes during pregnancy. As a result, fetal cardiac function can change due to fetal abnormalities such as hypoxia, hyperglycemia and volume or pressure overload [1]. Fetal cardiac function is therefore an interesting tool to assess the adaptation of the fetus to various pregnancy complications. It would be valuable to add to the current echocardiographic assessment which is mainly focused on the detection of structural defects.

Two-dimensional speckle tracking echocardiography (2D-STE) is an emerging technique within the research field for assessing fetal cardiac function by quantifying myocardial deformation during one cardiac cycle [2–5]. Global longitudinal strain (GLS) measurements of myocardial deformation are the most commonly used 2D-STE parameters in the fetus. GLS represents the shortening of the myocardial wall during one heartbeat. Abnormal GLS values were shown in several pregnancy complications, including fetal growth restriction, hypertensive pregnancy disorders, maternal diabetes and twin-to-twin-transfusion syndrome [6–9]. Fetal GLS measurements may also be of value in the diagnosis and follow-up of congenital heart defects, for example in coarctation of the aorta where the prenatal detection rate is low, while the false positive rate is relatively high. Speckle tracking values are shown to increase the detection of coarctation when used complementary to conventional parameters [10–12]. Prenatal diagnosis of these defects enables timely, adequate treatment and therefore reduces morbidity and mortality [13, 14].

2D-STE uses computer-based algorithms to determine GLS, but the user still needs to place and modify the tracking lines. Therefore, 2D-STE is considered a user-independent technique [3]. New users must undergo sufficient training to achieve good learning curves before they can independently report GLS values. The learning curve of tracking the endocardial walls for offline GLS analysis with 2D-STE has been set for adults but not yet for fetuses [15]. Before introducing 2D-STE in obstetrics, it is important to understand the learning curve for fetal GLS with 2D-STE [3].

Therefore, this study aimed to investigate the individual learning curve of tracking the endocardial walls for fetal left ventricular (LV) and right ventricular (RV) GLS analysis after 2D-STE training by an expert. The learning curves of three trainees with different levels of fetal echocardiographic experience (Maternal Fetal Medicine fellow, OBGYN resident and medical student) were examined to investigate the influence of fetal echocardiographic experience.

## Methods

### Study design

This was a secondary, retrospective analysis of a longitudinal prospective study to examine the learning curve of fetal LV- and RV-GLS analysis with 2D-STE [16]. Anonymized data was accessed the 12[th] of May 2021 and used for this study. Therefore, a waiver for ethical approval has been granted by the Medical Ethics Committee of Máxima MC, Veldhoven, The Netherlands (N21.038).

### Trainees

The expert (N.v.O.), a Maternal-Fetal medicine specialist, has ten years of fetal echocardiographic experience and five years of fetal 2D-STE experience [6, 16, 17]. The three trainees were: (1) a Maternal-Fetal Medicine fellow (A.F.) with fetal echocardiographic experience, (2) an OBGYN resident (T.N.) with little fetal echocardiographic experience and (3) a medical student (Z.v.L.) without any fetal ultrasound experience. The expert's GLS measurements were considered the reference standard for the learning curves of the trainees. The fellow, resident and medical student performed GLS measurements with 2D-STE for the first time in this study. Permission for publication of the results was obtained from all trainees.

### Training protocol

At baseline, the trainees attended the same training by the expert and a researcher (C.d.V.). The training consisted of a presentation on (1) the background and technique of 2D-STE to measure fetal cardiac function with GLS, (2) basic principles of fetal LV and RV anatomy, (3) the reason for investigating the learning curve of GLS analysis and (4) a live demonstration of one measurement on the different steps of GLS analysis with vendor-specific 2D Cardiac Performance 1.2 software, developed by TomTec Imaging Systems GmbH (Munich, Germany) including tips and pitfalls in fetal GLS analysis. Subsequently, the trainees followed a hands-on training of two other measurements with the offline 2D-STE software under the supervision of the expert and the researcher, both experienced with fetal speckle tracking. Written guidelines on GLS analysis with the 2D-STE TomTec software and support by the researcher on buttonology and offline analysis were available on request during all 100 measurements.

### Study population and data acquisition

Every trainee analyzed the same 100 fetal heart clips in the same order of sequence. The 100 fetal heart clips used in this study for offline GLS analysis were retrieved from the database of a longitudinal prospective study (NL64999.015.18) that included 124 healthy fetuses [16]. We used the clips of this study that were obtained during the anomaly scan (between 18+0 and 22 +0 weeks of gestation). All women had an uncomplicated singleton pregnancy, and no structural anomalies were seen during the scan.

The fetal heart clips were made by a Maternal-Fetal Medicine specialist (the expert N.v.O.) following a strict protocol during the anomaly scan [17]. An Epiq W7 ultrasound system (Philips, 134 Eindhoven, The Netherlands) equipped with a 9-MHz linear transducer was used [18]. The highest possible frame rates were achieved by first reducing B-mode image depth and sector width before using the Zoom box to fill the complete image with a four-chamber view. Frame rates were above the recommended 80 frames per second [19, 20].

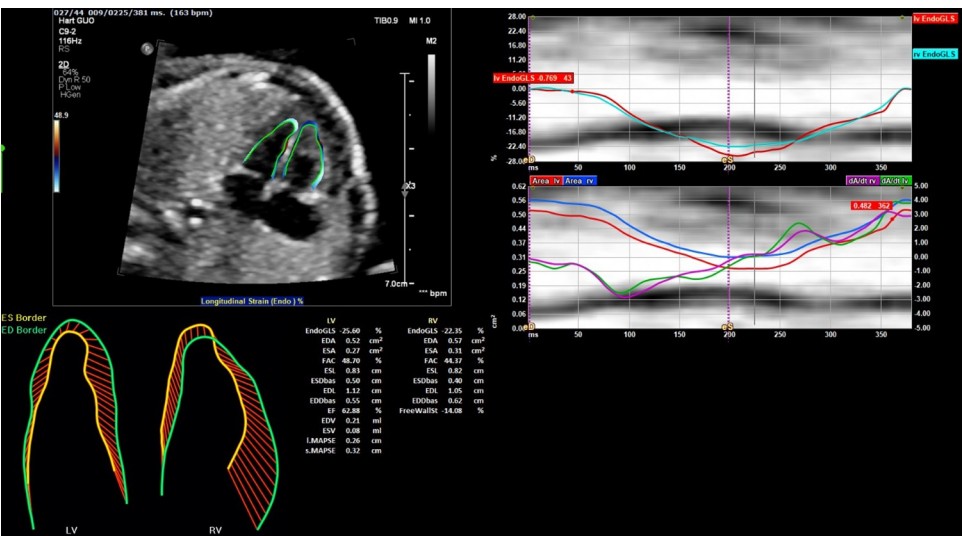

**Fig 1. GLS analysis with 2D-STE software.**

### 2D-STE offline analysis

The fetal heart clips were stored as DICOM clips and analyzed offline with the software program 2D Cardiac Performance 1.2 developed by TomTec Imaging Systems GmbH (Munich, Germany).

Fig 1 shows a GLS analysis with 2D-STE. The GLS analysis with the TomTec software consists of four steps: 1. one cardiac cycle, from an end-diastolic frame to the next end-diastolic frame, is isolated using the M-mode feature of the software, 2. the mitral annuli and the apex are identified with 3 index points in the end-systolic frame and the software tracks the endocardium in the end-systolic frame, which could be adjusted manually if needed. Step 3 the software tracks the endocardium in the end-diastolic frame which could be adjusted manually, and 4. the software calculates the GLS values. The expert performed all four steps. The trainees performed only the last three steps of GLS analysis using the same isolated cardiac cycle as the expert. All observers performed the measurements in the same order and were blinded to each other's results.

GLS was defined as the fractional change in myocardial length during one cardiac cycle (%) [2]. In the contracting heart, GLS represents the change from an end-diastolic length to an end-systolic length, i.e. the shortening of the cardiac wall, and has, therefore, a negative value. More shortening was referred to as a mathematically decreased GLS value (i.e. a more negative value).

### Statistical analysis

Statistical analyses were conducted with the statistical software package SPSS version 22 and the repeated measurement analysis with SAS version 9.4. Continuous data were presented by means with standard deviation (SD) or medians with interquartile ranges [IQR]. To explore the learning curves of the trainees, intra-class correlation coefficients (ICCs) and Bland-Altman plots were derived per quartile of 25 consecutively analyzed fetal heart clips by comparing the GLS values of the expert minus the different trainees. Every trainee analyzed the same 100 fetal heart clips in the same order. The order of the 100 clips was predetermined by a researcher (C.d.V.) by marking four quartiles of 25 clips that were comparable in difficulty

level to perform GLS analysis. This was done to avoid a higher ICC level being caused by a lower difficulty level rather than the trainees learning curve.

For computing ICCs, a two-way mixed-effects ANOVA model was used, considering single measurements and consistency. The learning curves (i.e. the ICC values on the y-axis for each consecutive group of 25 measurements on the x-axis) were plotted in two graphs, one for LV-GLS analysis and one for RV-GLS analysis. ICCs less than 0.5, between 0.5 and 0.75, between 0.75 and 0.9, and greater than 0.90 indicated poor, moderate, good, and excellent consistency, respectively [21]. To assess the learning curves and to compare curves between the different types of trainees a repeated measurements analysis was performed. More specifically, mixed effects models were fitted to the differences between the scores of the expert and trainees with time- and trainee-type dependent mean and variances for both the LV-GLS and RV-GLS outcome. Normality of the model residuals was verified. Effects on the mean of the score difference were tested using F-tests as implemented in PROC MIXED. Effects on the variance of the score difference were tested using likelihood ratio tests. This results in performing 10 (2x5) statistical tests. We mitigate the issue of multiple testing by applying a bonferroni correction to the p-values in the repeated measurements analysis, so that $p_{adj} = \min(p \cdot 10, 1)$.

## Results

### Study characteristics

The median gestational age during the ultrasound scan was 19+3 [19+1 to 19+5] weeks, and the maternal body mass index at inclusion was 22.4 [20.9 to 25.8] kg/m2. The median fetal heart rate was 150 [140 to 157] beats per minute. The frame rates were 115 (±23) frames per second or 47 [39 to 53] frames per cardiac cycle above the suggested frame rate [19].

### Primary outcomes

**Learning curve of LV-GLS analysis.**   Table 1 reports the means ± SD of the trainees and corresponding ICCs per 25 clips between the trainees and the expert for LV-GLS analysis. The ICCs are also presented in Fig 2. The fellow started with poor consistency (95% CI: 0.00–0.59) after 25 measurements, poor-to-moderate consistency (95% CI: 0.23–0.77) after 50 measurements and stable moderate-to-good consistency after 75 (95% CI: 0.48 to 0.88) and 100 (95% CI: 0.44 to 0.86) measurements respectively. The resident showed poor-to-moderate consistency after 25 (95% CI: 0.20 to 0.77), 50 (95% CI: 0.27 to 0.80), and 75 (95% CI: 0.34 to 0.83) measurements respectively, before increasing to moderate-to-good consistency after 100 measurements (95% CI: 0.51 to 0.88). The student showed poor-to-moderate consistency after 25 (95% CI: 0.17 to 0.76) and 50 (95% CI: 0.16 to 0.75) measurements, followed by moderate-to-good consistency (95% CI: 0.54 to 0.90) after 75 measurements and 100 (95% CI: 0.48 to 0.87) measurements respectively.

**Table 1. Analysis of correlation of LV-GLS over consecutive quartiles of 25 measurements.**

| Cases | Expert | Fellow | | Resident | | Student | |
|---|---|---|---|---|---|---|---|
| | Mean ± SD | Mean ± SD | ICC (95%-CI) | Mean ± SD | ICC (95%-CI) | Mean ± SD | ICC (95%-CI) |
| 1–25 | -22.85 ± 8.82 | -21.81 ± 7.67 | 0.27 (0.00 to 0.59) | -23.32 ± 7.82 | 0.55 (0.20 to 0.77) | -21.15 ± 7.67 | 0.53 (0.17 to 0.76) |
| 26–50 | -24.87 ± 8.50 | -28.68 ± 11.18 | 0.57 (0.23 to 0.77) | -22.04 ± 11.95 | 0.59 (0.27 to 0.80) | -19.64 ± 9.01 | 0.51 (0.16 to 0.75) |
| 51–75 | -24.09 ± 8.38 | -24.49 ± 9.00 | 0.74 (0.48 to 0.88) | -21.38 ± 9.54 | 0.65 (0.34 to 0.83) | -20.69 ± 7.34 | 0.77 (0.54 to 0.90) |
| 76–100 | -22.25 ± 6.58 | -23.61 ± 8.55 | 0.71 (0.44 to 0.86) | -17.36 ± 6.95 | 0.75 (0.51 to 0.88) | -17.39 ± 5.96 | 0.73 (0.48 to 0.87) |

CI = confidence intervals; GLS = global longitudinal strain; ICC = intra class coefficient; LV = left ventricular; SD = standard deviation.

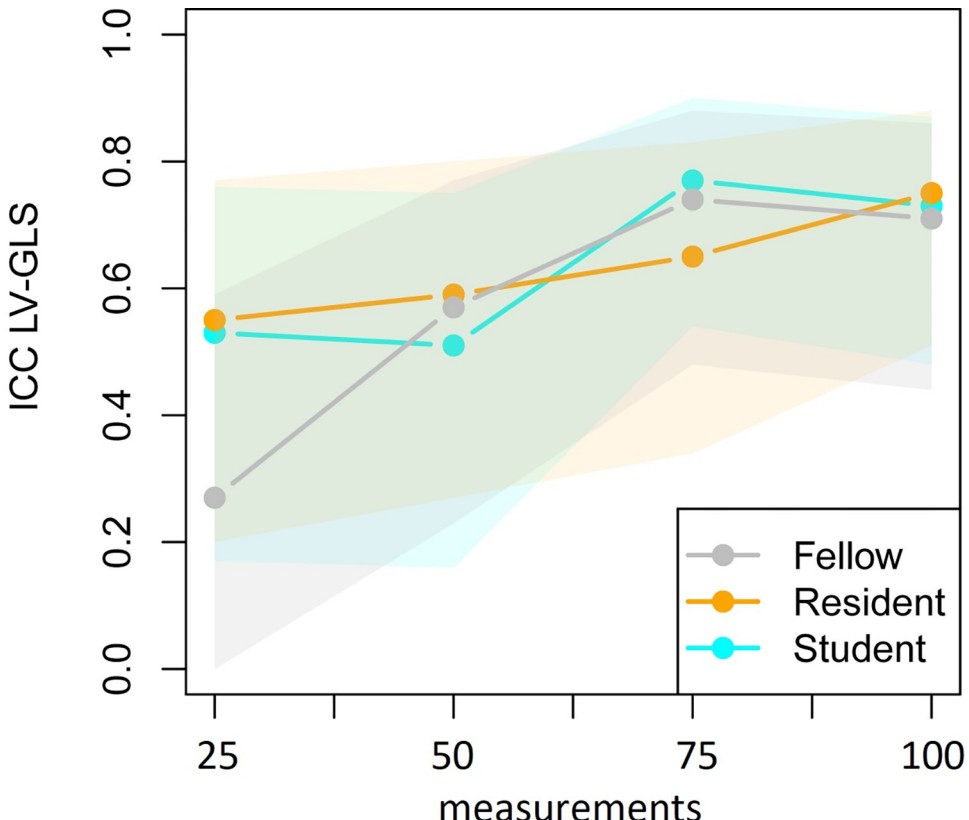

**Fig 2. ICC of each trainee LV-GLS value compared with the expert in four consecutive groups of 25 fetal heart clips.**

The mean and standard deviation of the difference in score of the expert and the trainees are presented in Table 2 and their development over time is shown in Figs 3 and 4 to present the LV-GLS learning curves. Based on the repeated measurements analysis the apparent decrease in variability of the differences between the score of the expert and trainees as presented in Fig 4 was found significant ($p_{adj} < 0.001$). The changes in variability were not found to be significantly different between the trainees ($p_{adj} = 1.00$). The mean of the difference

**Table 2. LV-GLS agreement analysis between expert and trainees over consecutive quartiles of 25 measurements.**

| | Fellow | | | | Resident | | | | Student | | | |
|---|---|---|---|---|---|---|---|---|---|---|---|---|
| **Cases** | **1–25** | **26–50** | **51–75** | **76–100** | **1–25** | **26–50** | **51–75** | **76–100** | **1–25** | **26–50** | **51–75** | **76–100** |
| **Difference expert and trainee** | -1,04 | 3,81 | 0,40 | 1,36 | 0,47 | -2,83 | -2,70 | -4,89 | -1,70 | -5,23 | -3,29 | -4,85 |
| **Upper CI mean difference** | 2,89 | 7,43 | 2,88 | 3,65 | 3,58 | 0,83 | 0,24 | -3,01 | 1,45 | -1,85 | -1,19 | -3,06 |
| **Lower CI mean difference** | -4,97 | 0,19 | -2,08 | -0,93 | -2,65 | -6,50 | -5,65 | -6,77 | -4,86 | -8,62 | -5,39 | -6,65 |
| **SD difference** | 10,02 | 9,24 | 6,33 | 5,84 | 7,94 | 9,34 | 7,51 | 4,79 | 8,05 | 8,64 | 5,36 | 4,59 |
| **Upper CI SD difference[a]** | 14,11 | 11,79 | 8,86 | 7,26 | 10,34 | 15,02 | 10,58 | 6,66 | 10,64 | 11,00 | 9,55 | 6,08 |
| **Lower CI SD difference[a]** | 7,44 | 7,24 | 4,78 | 4,68 | 6,34 | 6,29 | 5,49 | 3,74 | 6,09 | 6,55 | 3,34 | 3,70 |
| **Upper LoA** | 18,60 | 21,92 | 12,81 | 12,80 | 16,04 | 15,48 | 12,02 | 4,50 | 14,06 | 11,70 | 7,21 | 4,13 |
| **Lower LoA** | -20,69 | -14,30 | -12,00 | -10,09 | -15,10 | -21,15 | -17,42 | -14,28 | -17,47 | -22,17 | -13,79 | -13,84 |

CI = confidence intervals, LoA = Limits of Agreement. [a]obtained with bootstrapping

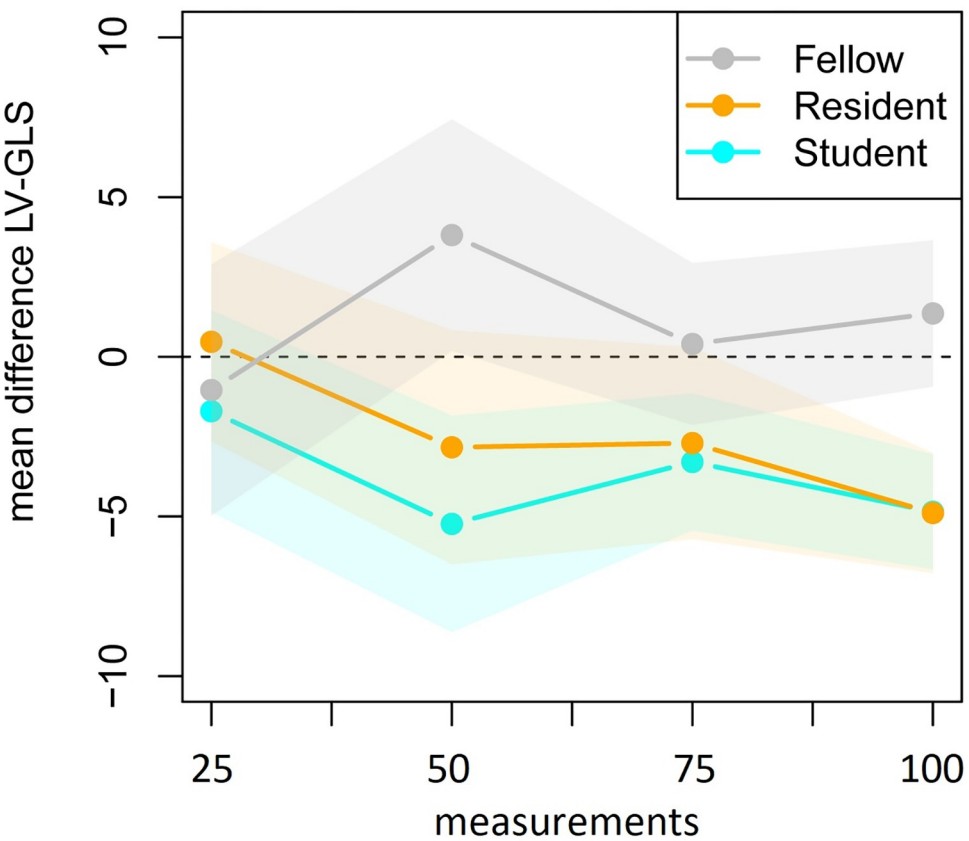

**Fig 3. Mean of the difference between the LV-GLS value of the expert and trainees in four consecutive groups of 25 fetal heart clips.**

between the expert and the trainees as presented in Fig 3 did not significantly change over time ($p_{adj}$ = 1.000), but the vertical positions of the curves were found to be significantly different between trainees ($p_{adj}$<0.001). The model residuals closely conformed to a normal distribution.

**Learning curve of RV-GLS analysis.** Table 3 reports the means ± SD of the novice trainees and corresponding ICCs per 25 clips between the expert and trainees for RV-GLS analysis. The ICCs are also presented in Fig 5. The fellow showed poor consistency after 25 (95%-CI 0.00–0.41) and 50 measurements (95%-CI 0.00–0.62), poor-to-moderate consistency after 75 measurements (95%-CI 0.20–0.77) and moderate-to-good consistency after 100 (95%-CI 0.39–0.84) measurements. The resident showed poor-to-moderate consistency after the first 75 measurements (95%-CI 0.32–0.82, 0.36–0.83 and 0.21–077 respectively) before evolving to moderate-to-good consistency after 100 (95%-CI 0.44–0.88) measurements. The student showed poor consistency (95%-CI 0.00–0.35) after 25 measurements, moderate-to-good consistency (95%-CI 0.51–0.88) after 50 measurements and poor-to-moderate realiability after 75 (95%-CI 0.00–0.67) and 100 measurements (95%-CI 0.00–0.66).

The mean and standard deviation of the difference in score of the expert and the trainees are presented in Table 4 and their development over time is shown in Figs 6 and 7 to represent the RV-GLS learning curves. Based on the repeated measurements analysis the apparent decrease in variability as presented in Fig 7 was found significant ($p_{adj}$<0.001). The changes in variability were not found to be significantly different between the trainees after applying the

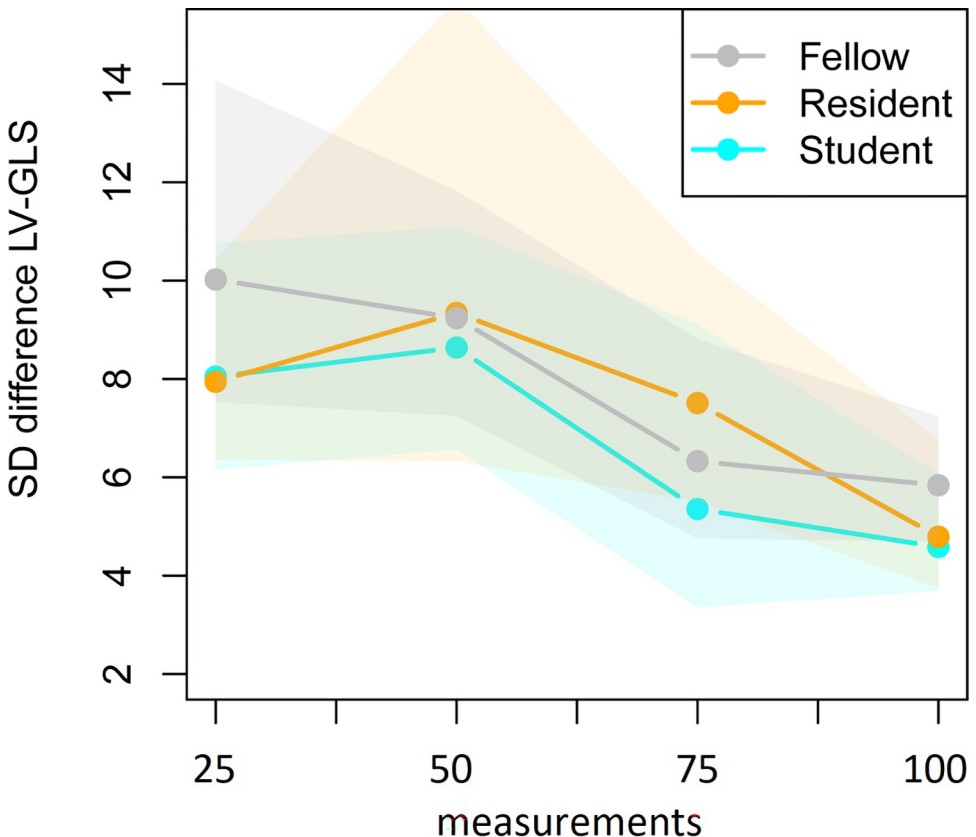

**Fig 4. Standard deviation of the difference between the LV-GLS value of the expert and trainees in four consecutive groups of 25 fetal heart clips.** Note: with corresponding 95% confidence intervals obtained via bootstrapping.

Bonferonni correction ($p_{adj}$ = 0.103). Also, the mean of the difference between the expert and the trainees, as presented in Fig 6, did significantly change over time ($p_{adj}$<0.001), but the shape of the change was not found to differ significantly between the trainees ($p_{adj}$ = 0.001). Finally, the vertical positions of the curves were found to be significantly different between trainees ($p_{adj}$ = 0.029). The model residuals closely conformed to a normal distribution.

**Bland Altman plots.** The Bland-Altman plots are visualized in the plots in S1 Appendix and support the findings of the repeated measurements analysis. The plots show acceptable limits of agreement, which become smaller over time for all three trainees.

**Table 3. Analysis of correlation of RV-GLS over consecutive quartiles of 25 measurements.**

| | Expert | Fellow | | Resident | | Student | |
|---|---|---|---|---|---|---|---|
| Cases | Mean ± SD | Mean ± SD | ICC (95%-CI) | Mean ± SD | ICC (95%-CI) | Mean ± SD | ICC (95%-CI) |
| 1–25 | -20.63 ± 7.01 | -22.27 ± 5.39 | 0.03 (0.00 to 0.41) | -22.61 ± 6.45 | 0.63 (0.32 to 0.82) | -19.47 ± 5.83 | 0.00 (0.00 to 0.35) |
| 26–50 | -20.35 ± 6.03 | -23.32 ± 9.44 | 0.30 (0.00 to 0.62) | -22.29 ± 8.57 | 0.65 (0.36 to 0.83) | -20.31 ± 7.74 | 0.75 (0.51 to 0.88) |
| 51–75 | -19.24 ± 5.64 | -20.69 ± 7.15 | 0.54 (0.20 to 0.77) | -18.73 ± 5.88 | 0.55 (0.21 to 0.77) | -18.10 ± 6.23 | 0.39 (0.00 to 0.67) |
| 76–100 | -22.76 ± 4.70 | -21.13 ± 5.56 | 0.68 (0.39 to 0.84) | -21.10 ± 5.70 | 0.71 (0.44 to 0.86) | -17.24 ± 5.08 | 0.36 (0.00 to 0.66) |

CI = confidence intervals; GLS = global longitudinal strain; ICC = intra class coefficient;. RV = right ventricular; SD = standard deviation

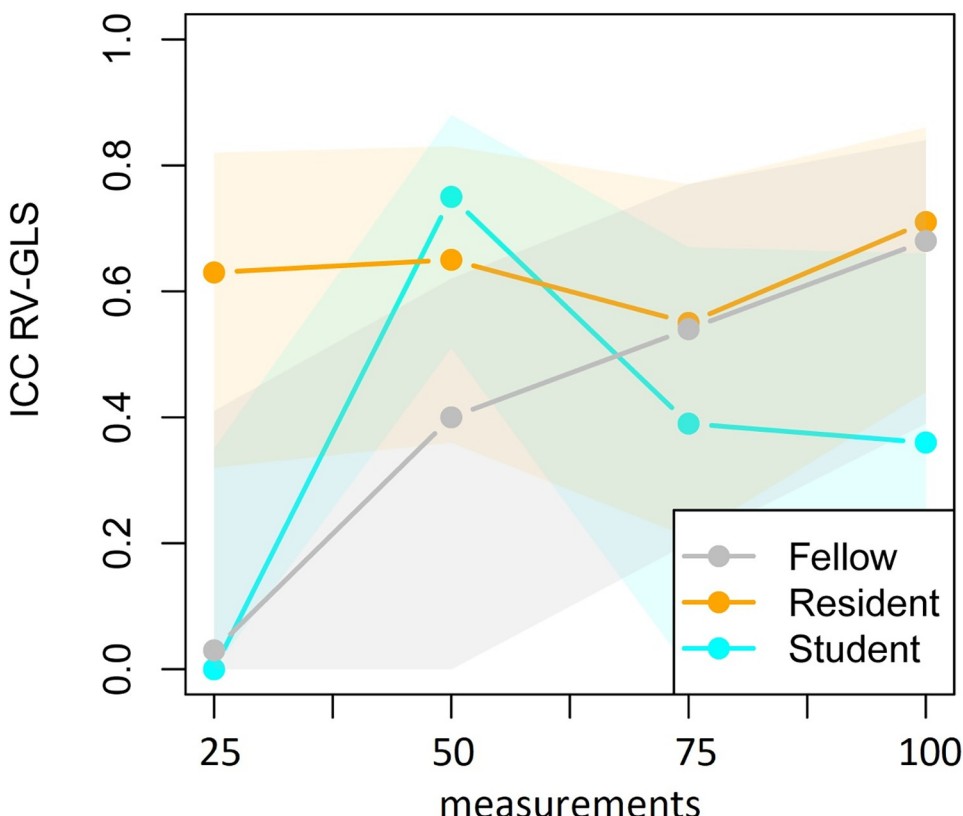

**Fig 5. ICC of each trainee RV-GLS value compared with the expert in four consecutive groups of 25 fetal heart clips.**

## Discussion

The consistency of reporting fetal GLS values in three trainees after 2D-STE training was moderate to good, except for RV-GLS analysis by the student, which was poor-to-moderate. As the number of measurements rised, all three trainees showed statistically significant progressive learning curves for LV- and RV-GLS analysis. The decrease of the variability of the difference in score with the expert over time was comparable between trainees for LV-GLS and RV-GLS.

**Table 4. RV-GLS agreement analysis between expert and novice trainees over consecutive quartiles of 25 measurements.**

|  | Fellow | | | | Resident | | | | Student | | | |
|---|---|---|---|---|---|---|---|---|---|---|---|---|
| **Cases** | **1–25** | **26–50** | **51–75** | **76–100** | **1–25** | **26–50** | **51–75** | **76–100** | **1–25** | **26–50** | **51–75** | **76–100** |
| **Difference expert and trainee** | 1,64 | 2,97 | 1,45 | -1,64 | 1,98 | 1,94 | -0,51 | -1,66 | -1,16 | -0,41 | -1,13 | -5,53 |
| **Upper CI mean difference** | 5,05 | 6,64 | 3,86 | -0,01 | 4,26 | 4,35 | 1,63 | -0,09 | 2,49 | 1,53 | 1,44 | -3,36 |
| **Lower CI mean difference** | -1,77 | -0,69 | -0,96 | -3,27 | -0,29 | -0,48 | -2,65 | -3,23 | -4,82 | -2,34 | -3,71 | -7,70 |
| **SD difference** | 8,71 | 9,36 | 6,15 | 4,15 | 5,80 | 6,17 | 5,45 | 4,01 | 9,33 | 4,93 | 6,58 | 5,53 |
| **Upper CI SD difference[a]** | 13,05 | 12,15 | 7,50 | 5,60 | 8,23 | 7,79 | 7,69 | 4,97 | 11,53 | 6,29 | 8,43 | 6,77 |
| **Lower CI SD difference[a]** | 6,08 | 7,05 | 4,90 | 3,28 | 4,18 | 5,13 | 3,77 | 3,40 | 7,63 | 3,92 | 5,40 | 4,60 |
| **Upper LoA** | 18,71 | 21,31 | 13,50 | 6,50 | 13,35 | 14,03 | 10,18 | 6,19 | 17,12 | 9,26 | 11,76 | 5,31 |
| **Lower LoA** | -15,42 | -15,36 | -10,60 | -9,78 | -9,38 | -10,16 | -11,20 | -9,51 | -19,45 | -10,08 | -14,03 | -16,37 |

CI = confidence intervals, LoA = Limits of Agreement. [a]obtained with bootstrapping

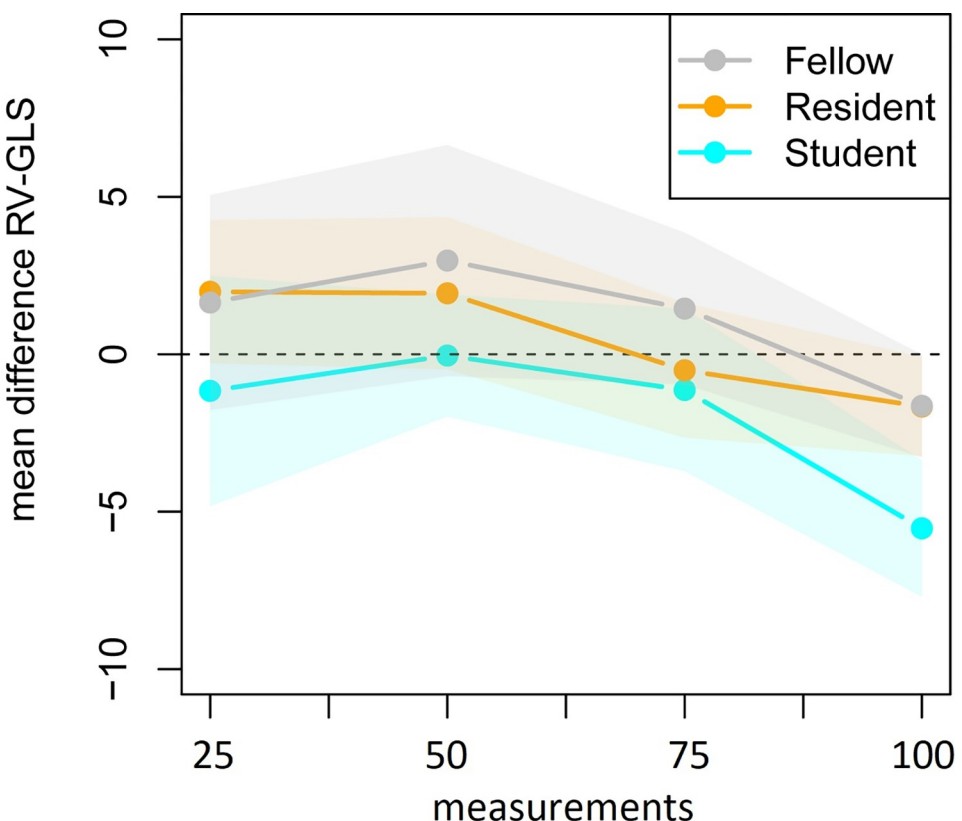

**Fig 6. Mean of the difference between the RV-GLS value of the expert and trainees in four consecutive groups of 25 fetal heart clips.**

The mean of the difference (score of the expert minus score of the trainee) did significantly differ between trainees for LV-GLS and RV-GLS.

The discrepancy among the trainees in terms of mean differences from the expert can be explained by different levels of fetal echocardiographic. Unexpectedly, the consistency of the resident and student was comparable to that of the fellow after 100 measurements (with the exception of the student's RV-GLS). Nevertheless, the fellow showed the lowest systematic error (mean difference from the expert) which likely reflects her experience with fetal heart ultrasound. The student's less reliable RV-GLS analysis after 100 measurements could be explained by the student's limited exposure to fetal heart ultrasound. Manual delineation of the RV wall is more challenging than the LV wall due to anatomical structures such as the moderator band.

In contrast to adult LV-GLS learning curves, this study did not reach excellent consistency (ICC≥0.90) for fetal GLS. The study conducted by Chan et al. showed excellent consistency while examining the learning curve of tracking the endocardial walls for LV-GLS analysis on 100 adult heart clips with 2D-STE across three groups of trainees with varying levels of echocardiographic experience (cardiology fellows, cardiac sonographers and medical students) [15]. However, the medical students showed suboptimal inter-observer reproducibility, also due to limited echocardiographic experience.

There are two possible reasons for the lack of expert competency in fetal GLS learning curves: firstly, the relatively lower reproducibility of fetal 2D-STE compared to adult STE, and

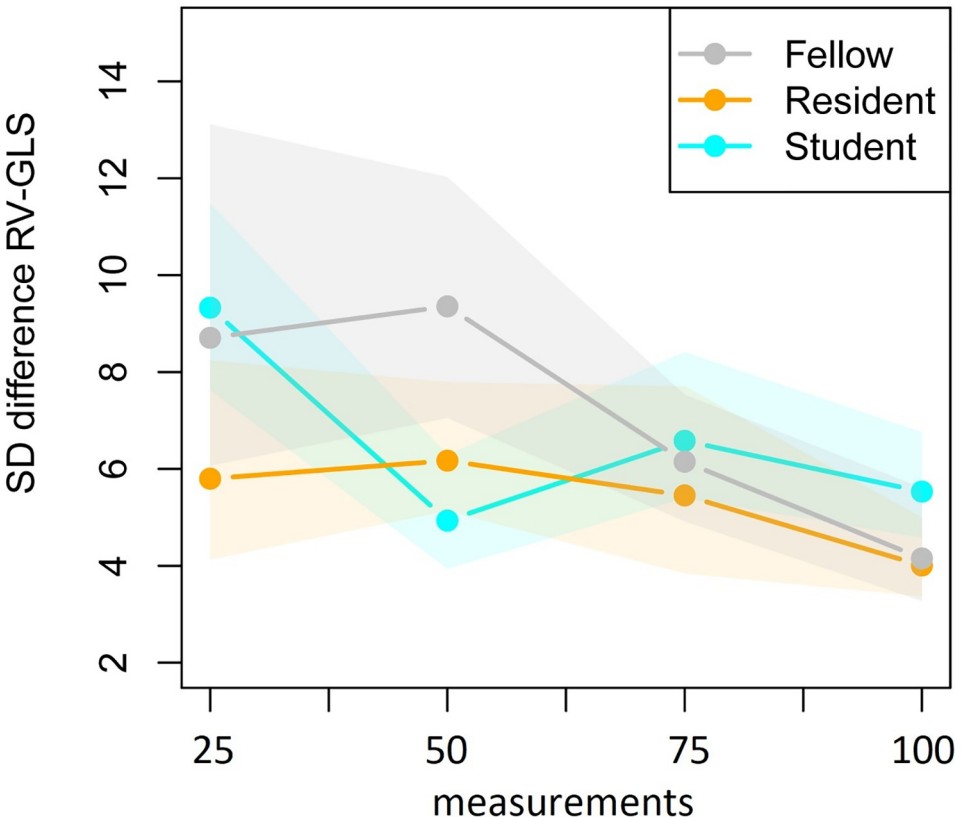

**Fig 7. Standard deviation of the difference between the RV-GLS value of the expert and trainees in four consecutive groups of 25 fetal heart clips.** Note: with corresponding 95% confidence intervals obtained via bootstrapping.

secondly, the training approach in the current study which was the adopted from Chan et al [15].

The ICCs observed in this study align with the reported inter-observer reproducibility of fetal GLS analysis in recent literature [22]. The learning curves were limited by these relatively low reproducibility values of fetal GLS analysis, given that the current inter-observer reproducibility for GLS analysis is much lower in fetuses (LV-GLS ICC 0.65 (95%-CI 0.43–0.79); RV-GLS 0.58 (95%-CI 0.32–0.74) than in adults (LV-GLS ICC 0.90 (95%-CI 0.81–0.94) [22, 23]. Speckle tracking might pose greater challenges in fetuses compared to adults due to various factors such as: (1) the greater distance from the ultrasound probe to the fetal heart, (2) the smaller size of the fetal heart is smaller and the higher heart rate, (3) influencing maternal factors such as obesity [24] and (4) variation in the insonation angle, due to fetal movements, whereas the position of the apex relative to the ultrasound beam remains constant in adults [25].

The second explanation could be that the training might have been too minimal to achieve an excellent learning curve compared to adult GLS learning curves, suggesting that more training measurements are required for fetal GLS analysis. The training in this study was based on the method by Chan et al. for adult LV-GLS analysis involving, two supervised measurements followed by 100 exercises using 2D-STE. To directly promote error correction, access to informative feedback by an experienced user of fetal speckle tracking on buttonology and offline analysis was available upon request throughout the 100 measurements. To ensure successful

training for new users, it is suggested to incorporate the elements of deliberate practice [26, 27]. The duration of learning may vary for each individual. Training should be continued until the trainees reach the expert level rather than being based on a fixed number of training measurements [26, 28]. The speckle tracking technique should be further improved to reduce user-dependency. Once improvements are made, the learning curve of strain analysis with speckle tracking could be re-investigated, potentially requiring fewer training measurements. In addition, this study only investigated individual learning curves. To determine the number of training measurements needed before strain can be reported independently, the learning curve should be assessed in a larger group of at least ten trainees after the technique is further improved.

## Future recommendations to improve fetal GLS reproducibility

There are different strategies to minimize inter-observer variability, thereby making fetal 2D-STE more reliable and applicable in the future. These strategies include: (1) integrating real-time fetal ECG which can lead to more precise identification of a cardiac cycle compared to the software's M-mode function, (2) minimizing manual adjustment of the tracking lines by refining the automatic segmentation within the 2D-STE software. Options to minimize manual adjustment are, for example, by automatizing the segmentation process through machine learning models or by exploring approaches such as combining different tracking methods, i.e. speckle tracking and contour tracking, to resolve issues of discontinuity with a single method [29–31]; (3) establishing international consensus on data acquisition and 2D-STE software for fetal application. The implementation of international standardization for data acquisistion and speckle tracking analysis significantly contributed to more accurate adult GLS analysis [18].

## Strenghts and limitations

This is the first study on the learning curve of offline fetal strain analysis and, therefore, provides insights how to improve this technique for fetal application. All trainees analyzed the same fetal heart clips, made by one experienced sonographer, to ensure high image quality which is essential for tracking quality. This is a strength and limitation of the study because acquisition settings have an important impact on the measurements. These settings include frame rates, and image quality and resolution which are dependent on the operator's experience and ultrasound system and transducer used [32–34]. For this study, the same image acquisition settings and same operator were used. If image acquisition variability by different trainees was included in the learning curve, this would complicate the determination of whether the variation in GLS measurements between trainees was due to image acquisition or offline GLS analysis.

An additional limitation is the generalizability of our study. We chose the TomTec software because this software is indepent from an ultrasound system. However, four other 2D-STE software programs are commonly used (GE, Philips, Siemens, Toshiba). We cannot extend our conclusions to these software because they use different buttonology, workflow processes and algorithms.

## Conclusions

This is the first study to examine the learning curve of 2D-STE GLS analysis in fetuses. Three trainees with varying levels of fetal echocardiographic experience (fellow, resident and student) showed progressively learning curves over time. The consistency in reporting fetal LV- and RV-GLS was found to be moderate to good, except for the student's RV-GLS analysis, which

was poor. At the end of their learning curves, out of the three trainees, the fellow showed the closest resemblance to the expert's GLS values. Based on these findings, we recommend that GLS analysis should be performed by observers experienced with fetal echocardiography.

However, the learning curves were limited by the relatively low inter-observer reproducibility of speckle tracking in fetuses compared to adults and should be reexamined after the reproducibility has improved. Then, the number of measurements needed to achieve expert competency for fetal GLS analysis using 2D-STE, should be examined in a group of at least ten trainees experienced with fetal heart ultrasound.

## Supporting information

**S1 Appendix. Bland Altman plots.**
(DOCX)

**S1 File.**
(SAV)

## Acknowledgments

We thank Marta Regis of the department of mathematics and computer science at the Technical University Eindhoven, the Netherlands, for her contribution to the statistical ICC analysis during the manuscript preparation.

We thank Philips Healthcare (Eindhoven, The Netherlands) and TomTec Imaging Systems GmbH (Munich, Germany) for providing the software.

## Author Contributions

**Conceptualization:** Chantelle M. de Vet, Thomas J. Nichting, Daisy A. A. van der Woude, Myrthe van der Ven, S. Guid Oei, Noortje H. M. van Oostrum, Judith O. E. H. van Laar.

**Data curation:** Chantelle M. de Vet, Thomas J. Nichting, Annemarie F. Fransen, Zoé van Lier, Noortje H. M. van Oostrum.

**Formal analysis:** Chantelle M. de Vet, Thomas J. Nichting, Richard A. J. Post.

**Investigation:** Chantelle M. de Vet, Thomas J. Nichting, Annemarie F. Fransen, Zoé van Lier.

**Methodology:** Chantelle M. de Vet, Thomas J. Nichting, Daisy A. A. van der Woude, Myrthe van der Ven, Richard A. J. Post, S. Guid Oei, Noortje H. M. van Oostrum, Judith O. E. H. van Laar.

**Project administration:** Chantelle M. de Vet, Thomas J. Nichting.

**Supervision:** Daisy A. A. van der Woude, Myrthe van der Ven, S. Guid Oei, Noortje H. M. van Oostrum, Judith O. E. H. van Laar.

**Visualization:** Chantelle M. de Vet, Thomas J. Nichting, Richard A. J. Post, Judith O. E. H. van Laar.

**Writing – original draft:** Chantelle M. de Vet, Thomas J. Nichting, Richard A. J. Post.

**Writing – review & editing:** Chantelle M. de Vet, Thomas J. Nichting, Annemarie F. Fransen, Daisy A. A. van der Woude, Myrthe van der Ven, Richard A. J. Post, Zoé van Lier, S. Guid Oei, Noortje H. M. van Oostrum, Judith O. E. H. van Laar.

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
