## [Decision Letter · Decision Letter 0]

10 Jun 2024

PONE-D-23-44042Two-dimensional fetal speckle tracking; a learning curve study for offline strain analysisPLOS ONE

Dear Dr. de Vet,

Thank you for submitting your manuscript to PLOS ONE. After careful consideration, we feel that it has merit but does not fully meet PLOS ONE’s publication criteria as it currently stands. Therefore, we invite you to submit a revised version of the manuscript that addresses the points raised during the review process.

We look forward to receiving your revised manuscript.

Kind regards,

Kumaradevan Punithakumar

Academic Editor

PLOS ONE

Additional Editor Comments (if provided):

Reviewers' comments:

Reviewer's Responses to Questions

**Comments to the Author**

1. Is the manuscript technically sound, and do the data support the conclusions?

Reviewer #1: Yes

Reviewer #2: Yes

Reviewer #3: No

Reviewer #4: Yes

2. Has the statistical analysis been performed appropriately and rigorously? 

Reviewer #1: Yes

Reviewer #2: Yes

Reviewer #3: N/A

Reviewer #4: Yes

3. Have the authors made all data underlying the findings in their manuscript fully available?

Reviewer #1: Yes

Reviewer #2: Yes

Reviewer #3: Yes

Reviewer #4: Yes

4. Is the manuscript presented in an intelligible fashion and written in standard English?

Reviewer #1: Yes

Reviewer #2: Yes

Reviewer #3: Yes

Reviewer #4: Yes

5. Review Comments to the Author

Reviewer #1: 

I would suggest that in the results chapter of the abstract as the only revision to remove the numerical values.

The introduction is well written and concise.

Material and method adequately described. In figure 1 I suggest you to improve the image quality because it is very difficult to discern the obtained values and end-diastolic moments in M-mode.

The results are consistent and support the conclusions. As a small comment from what it seems from the results for examiners without domain expertise it would probably require more than 100 examinations analyzed to adequately interpret the speckle tracking volumes.

The discussions are consistent and point out very well all the intricacies of the method and the learning curve of this method.

As said the conclusions are adequate and supported by the results of the study.

Reviewer #2: The Authors analyzed the reproducibility and learning curve of fetal 2D-STE.

Their results highlight the importance of an adequate learning curve for avoiding mistakes in both the acquisition and the interpretation of fetal 2D-STE.

The Authors also suggested some strategies to minimize the inter-observer variability, such as the ECG synchronization and minimizing the manual adjustment of the tracking lines by refining the automatic segmentation within the 2D-STE software.

I agree with the Authors' findings.

I have only a minor suggestion for the Authors.

In the Limitations section, the Authors could summarize the main limitations of strain echocardiographic imaging, particularly its dependence on:

1) good image quality;

2) frame rates (generally, no less than 80 fps for fetal 2D-STE); the tracking quality becomes reduced when the frame rate is too low;

3) the operator’s experience;

4) loading conditions (pre- and after-load);

5) the ultrasound system employed for the analysis (inter-vendor variability).

The Authors could cite the following references: PMID: 25762560, PMID: 28528162 and PMID: 36142856.

Reviewer #3: Dear authors,

It is interesting to compare trainees' learning curves to those of an expert in fetal echocardiography, especially fetal global longitudinal strain. In general, this is a very well-written article and a good idea, but one uncorrectable flaw is present in the reviewer's opinion.

An article should inform us of some new ideas, and statistical methods should be used to show how it is replicable in a different population (in this case, trainees). Since there were only three probands compared to one expert, we cannot draw any conclusions and accept any recommendation based on these data even less. The reviewer suggests increasing the probands by at least ten in a group and then comparing the group variability to correct for inter-individual variability of the learning curves. After completion of the testing rounds, there may be an add-on evaluation to correct for intra-individual variability, which can be found even in experts. Then, the reviewer will find this manuscript worth publishing in Plos One and encourage the authors to resubmit it.

Reviewer #4: Fetal echocardiography, until recently, was used exclusively to identify cardiac malformations. In recent years, however, special attention has been paid to the evaluation of the cardiac ventricular function in the fetus. The assessment of fetal cardiac function is challenging, given the small size and motion of the fetal heart, the moving fetus and the negative influence of high maternal body mass index

Two-dimensional speckle tracking echocardiography (2D-STE) represents an innovative parameter for the quantification of ventricular function which has gained ground in recent years. The use of this method during the fetal period is developing. That is why I consider that such a fetal study is welcomed.

1. The introduction is of appropriate length. On the other hand, from my point of view, the information provided by the authors is rather general, I would prefer something more specific about fetal 2D-STE.

2. The chapter on the method used is very explicit and well written. I would like to congratulate the authors for the complex but at the same time easy-to-follow way of this chapter.

3. The obtained results bring important information in the field of fetal 2D-STE that can be used to facilitate the implementation of this method.

4. The statistical analysis used is appropriate for this research.

5. Although there are few studies that address this topic, the discussions in the article are well written.

6. The conclusions are relevant to this article.

7. Tables and figures are clear and informative.

In conclusion, I congratulate the authors for the work done, for the clarity of this research but also for the new information brought in the field of fetal 2D-STE

6. PLOS authors have the option to publish the peer review history of their article (what does this mean?). If published, this will include your full peer review and any attached files.

Reviewer #1: **Yes: **VLADUT SASARAN

Reviewer #2: No

Reviewer #3: No

Reviewer #4: **Yes: **Cerghit-Paler Andreea-Maria

---

## [Author Response · Author response to Decision Letter 0]

1 Aug 2024

5. Review Comments to the Author

Reviewer #1: 

I would suggest that in the results chapter of the abstract as the only revision to remove the numerical values.

The introduction is well written and concise.

Material and method adequately described. In figure 1 I suggest you to improve the image quality because it is very difficult to discern the obtained values and end-diastolic moments in M-mode.

The results are consistent and support the conclusions. As a small comment from what it seems from the results for examiners without domain expertise it would probably require more than 100 examinations analyzed to adequately interpret the speckle tracking volumes.

The discussions are consistent and point out very well all the intricacies of the method and the learning curve of this method.

As said the conclusions are adequate and supported by the results of the study.

Response to Reviewer #1

Thank you for your sincere feedback. 

In the results section of the abstract, we aimed to summarize the learning curves of the trainees. We chose to retain the numerical values because the learning curves are individual and differ for all three trainees. 

We have made an effort to improve the quality of Figure 1 and used the journal recommended Preflight Analysis and Conversion Engine (PACE) digital diagnostic tool which ensures that figures meet PLOS requirements. 

We fully agree with your comment; if one strives for excellent consistency of the trainee compared to the expert (according to the adult learning curve paper), there are more than 100 training measurements needed. Alternatively, the speckle tracking technique should be further improved to reduce user dependency. Then, the learning curve of strain analysis with speckle tracking could be re-investigated and there are may be fewer training measurements needed. We have amended the Discussion – page 19-20: The second explanation could be that the training might have been too minimal to achieve an excellent learning curve compared to adult GLS learning curves, suggesting that more training measurements are required for fetal GLS analysis. The training in this study was based on the method by Chan et al. for adult LV-GLS analysis involving, two supervised measurements followed by 100 exercises using 2D-STE. To directly promote error correction, access to informative feedback by an experienced user of fetal speckle tracking on buttonology and offline analysis was available upon request throughout the 100 measurements. To ensure successful training for new users, it is suggested to incorporate the elements of deliberate practice26,27. The duration of learning may vary for each individual. Training should be continued until the trainees reach the expert level rather than being based on a fixed number of training measurements 26,28. The speckle tracking technique should be further improved to reduce user-dependency. Once improvements are made, the learning curve of strain analysis with speckle tracking could be re-investigated, potentially requiring fewer training measurements. In addition, this study only investigated individual learning curves. To determine the number of training measurements needed before strain can be reported independently, the learning curve should be assessed in a larger group of at least ten trainees after the technique is further improved. 

Reviewer #2: 

The Authors analyzed the reproducibility and learning curve of fetal 2D-STE.

Their results highlight the importance of an adequate learning curve for avoiding mistakes in both the acquisition and the interpretation of fetal 2D-STE.

The Authors also suggested some strategies to minimize the inter-observer variability, such as the ECG synchronization and minimizing the manual adjustment of the tracking lines by refining the automatic segmentation within the 2D-STE software.

I agree with the Authors' findings.

I have only a minor suggestion for the Authors.

In the Limitations section, the Authors could summarize the main limitations of strain echocardiographic imaging, particularly its dependence on:

1) good image quality;

2) frame rates (generally, no less than 80 fps for fetal 2D-STE); the tracking quality becomes reduced when the frame rate is too low;

3) the operator’s experience;

4) loading conditions (pre- and after-load);

5) the ultrasound system employed for the analysis (inter-vendor variability).

The Authors could cite the following references: PMID: 25762560, PMID: 28528162 and PMID: 36142856.

Response to Reviewer #2

Thank you for your valuable feedback on the limitations section and the pointed out references. We have amended the text accordingly (See page 20-21 of the Revised manuscript with track changes): 

Discussion – page 20-21: This is a strength and limitation of the study because acquisition settings have an important impact on the measurements. These settings include frame rates, and image quality and resolution which are dependent on the operator’s experience and ultrasound system and transducer used32-34. For this study, the same image acquisition settings and same operator were used. If image acquisition variability by different trainees was included in the learning curve, this would complicate the determination of whether the variation in GLS measurements between trainees was due to image acquisition or offline GLS analysis. 

Reviewer #3: 

Dear authors,

It is interesting to compare trainees' learning curves to those of an expert in fetal echocardiography, especially fetal global longitudinal strain. In general, this is a very well-written article and a good idea, but one uncorrectable flaw is present in the reviewer's opinion.

An article should inform us of some new ideas, and statistical methods should be used to show how it is replicable in a different population (in this case, trainees). Since there were only three probands compared to one expert, we cannot draw any conclusions and accept any recommendation based on these data even less. The reviewer suggests increasing the probands by at least ten in a group and then comparing the group variability to correct for inter-individual variability of the learning curves. After completion of the testing rounds, there may be an add-on evaluation to correct for intra-individual variability, which can be found even in experts. Then, the reviewer will find this manuscript worth publishing in Plos One and encourage the authors to resubmit it.

Response to Reviewer #3

Thank you for your feedback. We agree with the Reviewer that more trainees per group would be of value in a follow-up study to determine the number of training measurements needed before one can report strain values independently. This is only the first study examining the learning curve of fetal strain analysis with two-dimensional speckle tracking. The aim of this study was to investigate the individual learning curves of strain analysis by three trainees with different levels of fetal echocardiographic experience. 

Speckle tracking is considered worldwide by experts to be a promising technique to measure fetal cardiac function in certain pregnancy diseases and congenital heart defects and there are many papers on speckle tracking. However, the authors have some concerns according to this technique; there are only a few papers on reproducibility and there is no literature on the learning curve of users new to speckle tracking. This paper highlights that there is a clear learning effect for strain analysis; the trainee experienced with fetal heart ultrasound (without any experience with speckle tracking) showing the closest resemblance to the expert. However, this paper also highlights that the technique is still limited by the relatively low inter-observer reproducibility. The authors believe that the technique has to be improved for fetal application first, before it can be implemented in clinical practice. The authors give recommendations on how to improve the technique and consistency among centers using speckle tracking. Before implementation, it is also needed to examine the number of training measurements needed for a new user before independently report strain values, according to the feedback of Reviewer 3. The authors have amended the text in the Discussion accordingly: 

Discussion – page 20: In addition, this study only investigated individual learning curves. To determine the number of training measurements needed before strain can be reported independently, the learning curve should be assessed in a larger group of at least ten trainees after the technique is further improved.

Discussion – page 22: However, the learning curves were limited by the relatively low inter-observer reproducibility of speckle tracking in fetuses compared to adults and should be reexamined after the reproducibility has improved. Then, the number of measurements needed to achieve expert competency for fetal GLS analysis using 2D-STE, should be examined in a group of at least ten trainees experienced with fetal heart ultrasound. 

Reviewer #4: 

Fetal echocardiography, until recently, was used exclusively to identify cardiac malformations. In recent years, however, special attention has been paid to the evaluation of the cardiac ventricular function in the fetus. The assessment of fetal cardiac function is challenging, given the small size and motion of the fetal heart, the moving fetus and the negative influence of high maternal body mass index

Two-dimensional speckle tracking echocardiography (2D-STE) represents an innovative parameter for the quantification of ventricular function which has gained ground in recent years. The use of this method during the fetal period is developing. That is why I consider that such a fetal study is welcomed. 

1. The introduction is of appropriate length. On the other hand, from my point of view, the information provided by the authors is rather general, I would prefer something more specific about fetal 2D-STE.

2. The chapter on the method used is very explicit and well written. I would like to congratulate the authors for the complex but at the same time easy-to-follow way of this chapter.

3. The obtained results bring important information in the field of fetal 2D-STE that can be used to facilitate the implementation of this method.

4. The statistical analysis used is appropriate for this research.

5. Although there are few studies that address this topic, the discussions in the article are well written.

6. The conclusions are relevant to this article.

7. Tables and figures are clear and informative.

In conclusion, I congratulate the authors for the work done, for the clarity of this research but also for the new information brought in the field of fetal 2D-STE

Response to Reviewer #4

Thank you for your kind words and feedback. The authors agree with the reviewer that speckle tracking echocardiography is a promising technique for measuring fetal cardiac function, but there are still challenges to overcome within this technique that have to be examined. 

In response to your comment 1, we have specified the example coarctation of the aorta. The authors believe that speckle tracking can significantly improve the detection of this congenital heart disease. Please see our amendments in the Introduction (page 4): Fetal GLS measurements may also be of value in the diagnosis and follow-up of congenital heart defects, for example in coarctation of the aorta where the prenatal detection rate is low, while the false positive rate is relatively high. Speckle tracking values are shown to increase the detection of coarctation when used complementary to conventional parameters10–12. Prenatal diagnosis of these defects enables timely, adequate treatment and therefore reduces morbidity and mortality13,14.

---

## [Decision Letter · Decision Letter 1]

29 Aug 2024

Two-dimensional fetal speckle tracking; a learning curve study for offline strain analysis

PONE-D-23-44042R1

Dear Dr. de Vet,

We’re pleased to inform you that your manuscript has been judged scientifically suitable for publication and will be formally accepted for publication once it meets all outstanding technical requirements.

Kind regards,

Kumaradevan Punithakumar

Academic Editor

PLOS ONE

Additional Editor Comments (optional):

Reviewers' comments:

Reviewer's Responses to Questions

**Comments to the Author**

1. If the authors have adequately addressed your comments raised in a previous round of review and you feel that this manuscript is now acceptable for publication, you may indicate that here to bypass the “Comments to the Author” section, enter your conflict of interest statement in the “Confidential to Editor” section, and submit your "Accept" recommendation.

Reviewer #1: All comments have been addressed

Reviewer #2: All comments have been addressed

Reviewer #4: All comments have been addressed

2. Is the manuscript technically sound, and do the data support the conclusions?

Reviewer #1: Yes

Reviewer #2: Yes

Reviewer #4: Yes

3. Has the statistical analysis been performed appropriately and rigorously? 

Reviewer #1: Yes

Reviewer #2: Yes

Reviewer #4: Yes

4. Have the authors made all data underlying the findings in their manuscript fully available?

Reviewer #1: Yes

Reviewer #2: Yes

Reviewer #4: Yes

5. Is the manuscript presented in an intelligible fashion and written in standard English?

Reviewer #1: Yes

Reviewer #2: Yes

Reviewer #4: Yes

6. Review Comments to the Author

Reviewer #1: (No Response)

Reviewer #2: The Authors satisfied my suggestions. In particular, they added a number of important technical limitations of strain echocardiographic imaging in the Limitations section.

Reviewer #4: (No Response)

7. PLOS authors have the option to publish the peer review history of their article (what does this mean?). If published, this will include your full peer review and any attached files.

Reviewer #1: No

Reviewer #2: No

Reviewer #4: **Yes: **Cerghit-Paler Andreea

---

## [Editor Report · Acceptance letter]

7 Nov 2024

PONE-D-23-44042R1 

PLOS ONE

Dear Dr. de Vet, 

I'm pleased to inform you that your manuscript has been deemed suitable for publication in PLOS ONE. Congratulations! Your manuscript is now being handed over to our production team.

Kind regards, 

on behalf of

Professor Kumaradevan Punithakumar 

Academic Editor

PLOS ONE